# Mapping Species Distributions of *Latoia consocia* Walker under Climate Change Using Current Geographical Presence Data and MAXENT (CMIP 6)

**DOI:** 10.3390/insects15100756

**Published:** 2024-09-29

**Authors:** Yuhan Wu, Danping Xu, Yaqin Peng, Zhihang Zhuo

**Affiliations:** College of Life Science, China West Normal University, Nanchong 637002, China; wuyuhan2580@163.com (Y.W.);

**Keywords:** *Latoia consocia* Walker, MaxEnt, climate change, geographical distribution

## Abstract

**Simple Summary:**

*Latoia consocia* Walker is an important herbivorous pest that has severely invaded North China in recent years. This study utilized the MaxEnt model system to investigate the impact of climate conditions on its distribution. The results show that the optimum temperature range of *L. consocia* is 5~14.5 °C, and the optimum annual precipitation range is 1980~2580 mm. With the warming of the climate, the potential habitat area of *L. consocia* will decrease in the future. These findings help to provide a scientific basis for pest control and inform scientific management strategies.

**Abstract:**

*Latoia consocia* Walker is an important phytophagous pest that has rapidly spread across North China in recent years, posing a severe threat to related plants. To study the impact of climatic conditions on its distribution and to predict its distribution under current and future climate conditions, the MaxEnt niche model and ArcGIS 10.8 software were used. The results showed that the MaxEnt model performs well in predicting the distribution of *L. consocia*, with an AUC value of 0.913. The annual precipitation (Bio12), the precipitation of the driest month (Bio14), the temperature annual range (Bio7), and the minimum temperature of the coldest month (Bio6) are key environmental factors affecting the potential distribution of *L. consocia*. Under current climate conditions, *L. consocia* has a highly suitable growth area of 2243 km^2^ in China, among which Taiwan has the largest high-suitable area with a total area of 1450 km^2^. With climate warming, the potential habitat area for *L. consocia* shows an overall decreasing trend in future. This work provides a scientific basis for research on pest control and ecological protection. A “graded response” detection and early warning system, as well as prevention and control strategies, can be developed for potentially suitable areas to effectively address this pest challenge.

## 1. Introduction

With global warming, the natural distribution ranges of many insects have been altered. Some insects originally adapted to specific climate conditions may migrate towards warmer regions, while populations of insects in high-altitude or polar regions may be threatened [1,2,3]. Research results indicate that meteorological factors significantly influence the distribution and growth of insects, with temperature and precipitation being the primary climatic factors affecting insect distribution. Under continued climate change in the future, insects may adjust their characteristics and physiological activities to adapt to environmental changes, ensuring the survival and reproduction of their populations. The adaptive strategy may involve adjustments in population distribution range, abundance, and behavioral patterns to cope with the new challenges posed by climate change [4,5], consequently resulting in changes in their geographical distribution range as well as community composition and structure [6,7]. The impacts of climate change on pests are multifaceted, directly or indirectly altering their distribution, reproduction, development, and level of damage to crops. Overall, climate change generally exacerbates the threat that pests pose to agriculture and ecosystems. As the climate warms further and extreme weather events increase, the threats pests pose to crops and ecosystems are likely to become more severe. For this reason, the management of agriculture and ecosystems needs to adapt to the new challenges posed by climate change. To address this, strategies must be developed by analyzing different scenarios of current and future climate developments. First, it is crucial to identify key environmental factors (e.g., temperature, precipitation, and soil type) that influence pest distribution. Second, by analyzing these environmental factors together, it is possible to predict the geographic distribution of suitable habitats for insects. Predictions should be used in conjunction with integrated pest management (IPM) strategies and continuous monitoring mechanisms to address these increasing risks.

Large-scale populations of insects cause significant economic losses to many crops, including fruit trees, willows trees, and rice. Herbivorous insects account for approximately 50% of all insects, and their large-scale reproduction and spread often lead to widespread pest outbreaks in crops, resulting in around 18% of global agricultural losses [8]. *Latoia consocia* Walker is a synonym of *Parasa consocia* Walker [9], and *L. consocia* (Lepidoptera: Limacodidae) is a foliar pest with a wide host-plant range, causing damage to fruit trees, forest trees, and other vegetation [10]. The pest is especially widespread in the North China region and is considered a significant pest in economic forestry [11,12,13]. The pest mainly feeds on the leaves and tender shoots of plants, causing perforations, notches, or the biting off of branches [14], which reduces photosynthesis, increases transpiration, severely affects tree vigor, and can even result in the widespread death of trees [15]. The larvae of the pest have a relatively long growth period, typically consisting of 6 to 8 instars, making them more vulnerable to predation by birds and other predators. The pests have evolved various defensive strategies to counteract these threats, including mechanisms such as camouflage, aposematic coloration, and toxic spines [16]. While most larvae of the pest have a relatively long growth period and possess thorns and toxic hairs on their body surfaces, there are certain differences among species in terms of thorn structure, attachment sites, and toxin composition [17]. Pests not only impact human health but also serve as significant threats in agriculture and forestry. They affect a wide range of fruit trees, including peaches, plums, apricots, and cherries [18], as well as mulberry trees, willows, balsam fir trees [19], and the newly discovered castor [20]. In recent years, with the adjustment of China’s agricultural industry structure, the planting area of economic and urban greening trees has continuously expanded. This has led to increased human contact with Lepidopteran larvae, which primarily feed on plants as important host organisms. Consequently, there has been a rise in issues such as dermatitis caused by human skin contact with caterpillar hairs. Among these, allergic dermatitis caused by Limacodidae larvae is the most common [21]. Currently, research on these pests primarily focuses on aspects such as biology, population spatial distribution, ecological niche, pathogenic mechanisms, and control techniques [22].

Species distribution models (SDMs) are predictive tools established based on the relationship between species presence and environmental variables and used to infer species distribution across different geographic locations [23]. SDMs have been widely applied in geography, including species potential distribution prediction, the conservation of rare fauna and flora, the prevention of biological invasions, and paleontology [24,25]. Currently, the most commonly used ecological models for species distribution prediction include the Genetic Algorithm for Rule-Set Prediction (GARP) [26], the Domain Environmental Envelope Model (domain) [27], the Bioclimatic Model (BIOCLIM) [28], Ecological Niche Factor Analysis (ENFA) [29], and the Maximum Entropy (MaxEnt) model, which are widely used ecological niche analysis methods both domestically and internationally [30].

The Maximum Entropy (MaxEnt) model, as a widely used species distribution model, has advantages such as short runtime, the ease of operation, good performance, and high accuracy [31,32], The MaxEnt model utilizes the principle of maximum entropy to combine known species distribution data and environmental factors, deducing the probability distribution of species distribution, thereby assessing and predicting the potential distribution range of species under different environmental conditions [33,34]. The MaxEnt model only requires species presence (or occurrence) data and environmental information, and it can effectively handle limited occurrence data with small sample sizes [35,36]. The MaxEnt model accepts species data as presence-only data without the need to consider missing data or excessive zero values, simplifying the data preparation process. The MaxEnt model can directly generate spatially explicit habitat suitability maps, visually displaying the probability of species distribution under different environmental conditions. It provides important references for ecological research and conservation decision-making. The receiver operating characteristic (ROC) curve method is a commonly used approach to assess the accuracy of the simulation results of the MaxEnt model. In this method, the area under the curve (AUC) is used as a metric to evaluate the predictive accuracy of the model. The AUC value ranges from 0 to 1, indicating the model’s classification ability and predictive performance. When the AUC value approaches 1, it indicates that the predictive performance of the model is better and the accuracy is higher [37].

Our study utilized an optimized MaxEnt model, combined with climate data and the latest global distribution data of the *L. consocia*, to predict its current and future distribution. By identifying the optimal parameter settings that influence model performance, the accuracy and reliability of the predictions were improved. The study analyzed the impact of major bioclimatic variables on the distribution of *L. consocia*, revealing its suitable habitat conditions. Based on current and future global climate scenarios, the study predicted the potential distribution of *L. consocia*, providing guidance for further risk assessment and monitoring efforts. The research offers significant theoretical support for the risk assessment of the global spread of *L. consocia*, as well as for precise monitoring and scientific control of this pest.

## 2. Materials and Methods

### 2.1. Species Occurrence Data

Biodiversity data for the *L. consocia* species used in this study were obtained by accessing the Global Biodiversity Information Facility (GBIF) (http://www.gbif.org, accessed on 13 May 2023). Data were also retrieved from publicly available journal articles related to the geographic distribution of *L. consocia* in domestic publications to obtain more comprehensive species distribution information. To ensure data accuracy and eliminate redundancy, duplicate records with the same sampling point and time were removed, and unclear geographical locations or abnormal specimen data were processed. Spatial filtering was conducted to ensure that each grid cell (10 km × 10 km) contained only one point. After these processing steps, the latitude and longitude coordinates of the samples were recorded in an Excel database and converted into CSV format for use in developing the MaxEnt model. These data processing steps help to improve data quality and accuracy, providing a reliable data foundation for subsequent model analysis. Figure 1 shows the habitus of the *L. consocia* collected from different regions. The occurrence map of *L. consocia* can be seen in Figure 2.

### 2.2. Bioclimatic Variables

In this study, 19 historical global bioclimatic variables were downloaded from the Global WorldClim database (http://www.worldclim.org/download, accessed on 10 June 2023). The environmental variable data were clipped and extracted using the Mask Extract tool in ArcGIS 10.4 to retain the required data range and then converted to ASCII format. Additionally, a provincial-level administrative map at a scale of 1:1.4 million was downloaded from the National Fundamental Geographic Information System of China (https://www.ngcc.cn/) as the analysis base map.

These bioclimatic variables closely affect the growth and development of *L. consocia*. Pearson correlation analysis was conducted in SPSS 26.0 to assess the linear correlation among climatic variables. Climatic variables with correlation coefficients greater than 0.85 were removed to reduce the influence of overfitting on the model, ensuring the robustness and accuracy of the model [38,39,40]. The distribution prediction model (Table 1) was constructed using four bioclimatic variables: minimum temperature of the coldest month (Bio6), temperature annual range (Bio5-Bio6) (Bio7), annual precipitation (Bio12), and precipitation of the driest month (Bio14).

### 2.3. Model Settings and Operation

To reduce the impact of spatial autocorrelation and sampling bias on the final predictions, the SDM Toolbox (version 2.5) was used to process the sparse occurrence data, ensuring that only one distribution point was included per 1 km × 1 km grid cell [41]. This sparse point method was aligned with the resolution of the environmental data to minimize the number of distribution points affected by spatial autocorrelation and to avoid model overfitting. The sorted and filtered data were imported into MaxEnt 3.4.1 software to prepare for the evaluation and model construction of bioclimatic variables. Jackknife tests were conducted to evaluate the bioclimatic variables. The obtained model randomly selected 25% of the species occurrence data as the test set, with the remaining 75% used as the training set. The model was evaluated and validated through 10 repetitions, up to 5000 iterations, and the default parameter selection using the Bootstrap method [42].

### 2.4. Model Evaluation

According to general understanding, different ranges of AUC values correspond to different levels of accuracy. AUC < 0.7 indicates low predictive accuracy and low reliability of the model; 0.7 ≤ AUC < 0.8 indicates moderate accuracy of the model, and its predictions can be used with caution. For 0.8 ≤ AUC < 0.9, it indicates a good accuracy of the model with a high reliability of the predictions, whereas AUC ≥ 0.9 indicates very accurate predictions by the model, suitable for further analysis. Therefore, evaluating the model through the ROC curve and AUC assessment can provide a comprehensive understanding of the model’s accuracy level, thus serving as an important reference for subsequent decision-making and analysis [43]. TSS values range from 0 to 1, with values greater than 0.7 indicating high predictive accuracy and values below 0.5 suggesting poor accuracy.

Convert MaxEnt output ASC files to raster format files in ArcGIS. According to the IPCC report on probability classification methods [44], set thresholds and colors for four categories of suitability levels. Height appropriate zone (*p* ≥ 0.66), set to 1, the color is red. Medium suitable area (0.33 ≤ *p* < 0.66), set to 2, the color is orange. Low suitability zone (0.05 ≤ *p* < 0.33), set to 3, the color is yellow. Inappropriate zone (*p* < 0.05), set to 4 and color to white [45,46].

## 3. Results

### 3.1. Key Environment Variable Selection and Model Performance

The MaxEnt software determines the importance of input variables in the final model by their contribution percentage. By inputting species occurrence data into the MaxEnt software for modeling and conducting jackknife tests, the contribution of each environmental variable can be obtained. The results indicate that annual precipitation (Bio12) contributes the most to model building, accounting for 74.8%, followed by precipitation of the driest month (Bio14) (13.41%), minimum temperature of the coldest month (Bio6) (9.22%) and temperature annual range (Bio7) (2.57%), respectively (Table 2). During the model construction process, interference with the model’s performance may occur if there is a high correlation between selected environmental variables. Therefore, it is important to exclude highly correlated variables when filtering variables. Ultimately, the minimum temperature of the coldest month (Bio6), temperature annual range (Bio7), annual precipitation (Bio12), and the precipitation of the driest month (Bio14), the four main environmental variables, were selected, as they effectively reflect the habitat potential of this species. It is noteworthy that the correlation coefficients among all selected variables are below 0.85, meeting the criteria for model selection and avoiding potential interference.

To critically assess the accuracy of the model in predicting the distribution of *L. consocia*, we conducted 10 runs using Chinese *L. consocia* distribution data and incorporating environmental variables to derive the KAPPA (0.78), TSS (0.89), and AUC (0.913) values for the MaxEnt model. According to Figure 3, the AUC value of the distribution model for *L. consocia* is 0.913, indicating that the MaxEnt model has high predictive accuracy, providing relevant data for studying the potential impact of climate change on the spread of *L. consocia*. As shown in Figure 4, the blue bars represent the importance of each variable in the species distribution, with the length of the bar positively correlated with the variable’s importance. The minimum temperature of the coldest month (Bio6), temperature annual range (Bio7), annual precipitation (Bio12), and the precipitation of the driest month (Bio14) were identified as important environmental variables affecting the distribution of *L. consocia*, with their training gain scores all exceeding 1.0. Among them, annual precipitation (Bio12) had the highest training gain score as the only variable, exceeding 1.5, indicating its important impact on the species distribution. On the other hand, the green bars represent the uniqueness of the variables, with the length of the bar negatively correlated with the amount of unique information contained in the variable. The green bar for the temperature annual range (Bio7) is the shortest, indicating that the temperature annual range (Bio7) contains important information not found in other environmental variables. Through these steps, four climatic environmental factors were determined, and the final distribution model for *L. consocia* was established.

### 3.2. Current Potential Distribution of L. consocia

The distribution prediction map of the optimal habitat for *L. consocia* was established using the MaxEnt model. The map was divided into four categories based on suitability level, highly suitable area, moderate suitable area, low suitable area, and unsuitable area, as shown in Figure 5. According to the simulation results, under current climatic conditions, the areas highly suitable for the growth of *L. consocia* are mainly concentrated in Taiwan Province, southeastern Tibet, and the coastal areas of Guangdong. Under the current climatic scenario, the potential distribution of *L. consocia* is mainly concentrated in the East China and Southwest China regions of China.

Statistics were conducted on the suitable habitat areas across 21 provinces and municipalities nationwide. Table 3 presents the statistical data for the main suitable distribution areas of *L. consocia*. The results indicate that the highly suitable areas for *L. consocia* are primarily concentrated in Taiwan (1450 km^2^), Tibet (472 km^2^), and Guangdong Province (261 km^2^), accounting for 4.03%, 0.04%, and 0.15% of the total highly suitable area in each province, respectively. The remaining highly suitable areas are distributed in Hong Kong, Guangxi, Yunnan, and Hainan, but with relatively smaller areas. It is worth noting that although the total area of highly suitable zones in Hong Kong is relatively small, they account for a higher percentage of the total area of the province. The total area of highly suitable zones nationwide is 2243 km^2^, accounting for 0.023% of the total national area.

### 3.3. Potential Distribution of L. consocia under Future Climatic Conditions

An analysis was conducted on the potential distribution of *L. consocia* under different climate change scenarios, and the results are shown in Figure 6 and Table 4. The suitable habitat for *L. consocia* is mainly distributed in the East China region, with a small portion found in the Southwest and South China regions (Figure 6). Under the SSP1-2.6 scenario in the 2050s, the highly suitable areas in Tibet, Guangxi, and Guangdong gradually increased but remained concentrated in Taiwan. Meanwhile, the moderately suitable areas in the Central China, Southwest, and East China regions decreased. Under the SSP2-4.5 scenario in the 2050s, the moderately suitable areas in the East China region slightly increased, especially in the South China region, particularly in Hainan Province. Under the SSP5-8.5 scenario in the 1950s, the highly suitable areas in Yunnan Province in the southwest and coastal areas in the South China region gradually increased, while the overall area of moderately and lowly suitable areas nationwide decreased. Under the SSP1-2.6 scenario in the 1990s, although the highly suitable areas remained concentrated in Taiwan and the highly suitable areas in Hainan Province increased, the highly suitable areas in Tibet and Guangxi decreased, and the moderately suitable areas in the East China and Central China regions decreased significantly. Under the SSP2-4.5 scenario in the 1990s, the areas of highly and moderately suitable habitat continued to decrease. Under the SSP5-8.5 scenario in the 1990s, the potential suitable habitat nationwide gradually decreased towards the southeast, but the highly suitable areas in the lower parts of the Southwest and South China regions increased. Overall, compared to current conditions, the growth area of suitable habitat for *L. consocia* is decreasing under the three emission scenarios in the future, indicating weaker adaptability of *L. consocia* to various climate changes in the future. This suggests that warming has a negative impact on the distribution of *L. consocia* within a certain temperature range.

From Table 4, it can be observed that under the SSP1-2.6 scenario, the total area of highly suitable habitat for *L. consocia* increased to 0.31 × 10^4^ km^2^ in the 2050s but decreased to 0.20 × 10^4^ km^2^ in the 2090s, with reductions in potentially suitable areas evident in both the 1950s and 2090s. Under the SSP2-4.5 scenario, both highly and moderately suitable areas continued to decrease by 60.41% and 34.20%, respectively. Overall, in the SSP1-2.6, SSP2-4.5, and SSP5-8.5 climate scenarios, the areas of moderately and lowly suitable habitats showed a decreasing trend. Additionally, under all three climate scenarios (SSP1-2.6, SSP2-4.5, and SSP5-8.5), the area of unsuitable habitat exhibited an increasing trend, except for a decrease in the SSP5-8.5 scenario in the 2050s. Specifically, under the SSP2-4.5 scenario, the area of highly suitable habitat for *L. consocia* decreased by 26.84% and 3.52%, respectively, while under the SSP5-8.5 scenario, the area of highly suitable habitat increased to 0.36 × 10^4^ km^2^ and 0.30 × 10^4^ km^2^, respectively.

### 3.4. Environmental Variables That Affect Geographic Distribution

The MaxEnt model has yielded satisfactory results in simulating the potential habitat of *L. consocia* in China. The AUC value of the test data is 0.913, which is higher than the AUC value of the random model (0.5). Jackknife tests of environmental factor importance in the MaxEnt model indicate that the minimum temperature of the coldest month (Bio6), temperature annual range (Bio7), annual orecipitation (Bio12), and the orecipitation of the driest month (Bio14) are the major contributing factors to the potential habitat of *L. consocia*.

The response curve reflects the quantitative relationship between the logical probability of species presence and environmental factors (Figure 7). Based on the response curve, the minimum temperature of bio6 follows a normal distribution, with the most suitable temperature ranging from 5 to 14.5 °C. For temperature annual range (Bio7), suitability is high below 21.4 °C. The optimum range of annual precipitation (Bio12) is from 1980 to 2580 mm. The optimum range of the precipitation of the driest month (Bio14) is 42–180 mm. Overall, both precipitation and temperature are important factors influencing the distribution probability of *L. consocia*.

## 4. Discussion

MaxEnt modeling is a niche model based on machine learning and mathematical statistics commonly used to predict species’ suitable habitats. This paper utilized the MaxEnt model to establish a potential distribution model for *L. consocia* and analyzed its distribution patterns to predict the potential impacts under current and future climate scenarios. The MaxEnt model demonstrated levels of “excellence” and “accuracy”, showing high AUC values and credibility [47]. The results indicate that the AUC value is 0.913, suggesting that the MaxEnt model performs well [48]. The MaxEnt modeling demonstrates reliability in predicting the habitat suitability of pest, providing crucial insights into the probability of pest distribution in specific regions. One of the contributions of this predictive model is to guide future field surveys, monitoring, and control efforts. Focused quarantine control studies in high-risk areas for target pests can expedite in-depth research processes, aiding in the development of more effective pest management strategies. The application of this model also enhances the understanding of pest ecology and behavior, providing scientific foundations for biodiversity conservation and agricultural production.

Under current climatic conditions, the suitable growth areas for *L. consocia* in China are mainly concentrated in Taiwan Province, southeastern Tibet, and the coastal areas of Guangdong Province. These regions exhibit high suitability for the growth of *L. consocia* due to their climatic conditions. According to the statistics of the suitable habitat area of 21 provinces and cities in China, the highly suitable areas for *L. consocia* growth are Taiwan (1450 km^2^), Xizang (472 km^2^), and Guangdong (261 km^2^). These statistical results demonstrate the potential distribution range of *L. consocia* in China, providing an important reference for further monitoring and management efforts. By analyzing the suitable habitat areas in different regions, we can better understand the distribution patterns of pests and take targeted control measures to protect the local ecological environment and crop safety.

Global warming will greatly affect species distribution by causing range expansion, migration, or contraction [49,50]. Under different climate change scenarios, the suitable habitat area and distribution of *L. consocia* exhibit complex patterns of change. Across various scenarios, there are increases or decreases in the area of highly suitable regions for *L. consocia*, mainly concentrated in the East China region, with fewer occurrences in the Southwest and South China regions. However, the overall trend indicates a potential decrease in suitable habitat areas for *L. consocia* in the future, suggesting a relatively weak adaptive capacity to future climate change. Across different scenarios, there are varying degrees of increase or decrease in the highly suitable habitat area for *L. consocia* in the 2050s and 2090s, while the area of moderately and lowly suitable habitats generally decreases. Under the SSP2-4.5 scenario, the highly and moderately suitable areas continue to decrease, while under the SSP5-8.5 scenario, the highly suitable area increases. Overall, under future climate change scenarios, the changes in suitable habitat areas for *L. consocia* are influenced by multiple factors, primarily temperature and precipitation climatic factors. Additionally, under different scenarios, there are trends of increasing or decreasing areas of unsuitable habitats, with a decrease in unsuitable areas observed under the SSP5-8.5 scenario. This indicates that *L. consocia* may face increasingly adverse ecological environments in the future, necessitating more ecological management and conservation efforts to mitigate its impacts.

Although the MaxEnt model performs well in predicting pest habitat suitability, it still has some limitations. The limited occurrence data used may lead to prediction bias, while the model struggles to comprehensively capture changes in pests under different environmental conditions [51]. The choice of environmental variables is critical to model results, and too many variables may introduce redundant information, increase complexity, and lead to unpredictable bias. In the study, 19 environmental factors were utilized to construct a distribution model, and, by eliminating variables with high correlation, annual precipitation, precipitation in the driest month, minimum temperature in the coldest month, and the annual range of temperature change were finally selected as key climatic variables to improve model accuracy [52]. However, the applicability of the model may be limited if factors such as soil type, topography, vegetation, and host plants are not considered. Considering a variety of environmental influences and comparing different types of models is critical to the impact of predicted outcomes. Incorporating more environmental factors into the model can provide a more comprehensive understanding and prediction of the distribution ranges of species, thus providing a more comprehensive and scientific basis and support for biodiversity conservation and management. Therefore, future studies can consider integrating multiple environmental factors and exploring the differences in the prediction results of different types of models to improve modeling accuracy and prediction ability. By integrating multiple environmental factors, we can better understand the ecological needs and adaptations of organisms, and thus more effectively guide the conservation and management of biodiversity [53,54].

While we can infer potential changes in species distribution through analysis of environmental and climate change, it does not imply the ability to provide accurate predictions. Assessing the impact of global climate change on species distribution is crucial. Taking correct and visionary actions can help us better understand the relationship between species and the environment, identify priority conservation areas, and formulate effective species conservation and resource utilization strategies. By analyzing the reasons for species distribution contraction, we can develop possible strategies to prevent, mitigate, or reverse negative trends, thus better protecting biodiversity and ecosystem stability [55]. Therefore, it is crucial to understand the trends in potential distribution changes of species like *L. consocia* and take corresponding conservation and management measures. Closely monitoring the impact of climate change on its habitat and developing corresponding adaptive control strategies are key to effective control of *L. consocia*.

## 5. Conclusions

This study integrated the MaxEnt model with ArcGIS technology to successfully calculate the current and potential suitable habitat distribution of *L. consocia* based on known distribution information and climatic factors. The results indicate that the AUC value of the MaxEnt distribution model is 0.913, demonstrating good performance in predicting the distribution of *L. consocia*. Both precipitation and temperature play important roles in shaping the suitable habitat for *L. consocia*. Under current climate conditions, the area highly suitable for *L. consocia*’s growth is 2243 km^2^, with Taiwan having the largest highly suitable area of 1450 km^2^, followed by Tibet and Guangdong Province. In the future climate, the potential habitat of *L. consocia* is expected to decrease towards the southeast. Additionally, warming had a negative effect on the distribution of *L. consocia* within a certain range, suggesting that rising temperatures would adversely affect the survival and reproduction of this species. By matching climate data with *L. consocia* distribution data and considering the uncertainty of future climate change, this study provides detailed information about potential habitats for *L. consocia*. This aids decision-makers in formulating more specific control strategies based on the research results and provides a theoretical basis for agricultural management. These research findings are of great significance for the development of effective management measures for *L. consocia*. They can help protect crops from the impact of this pest and maintain the stability and sustainable development of agricultural production.

## Figures and Tables

**Figure 1 insects-15-00756-f001:**
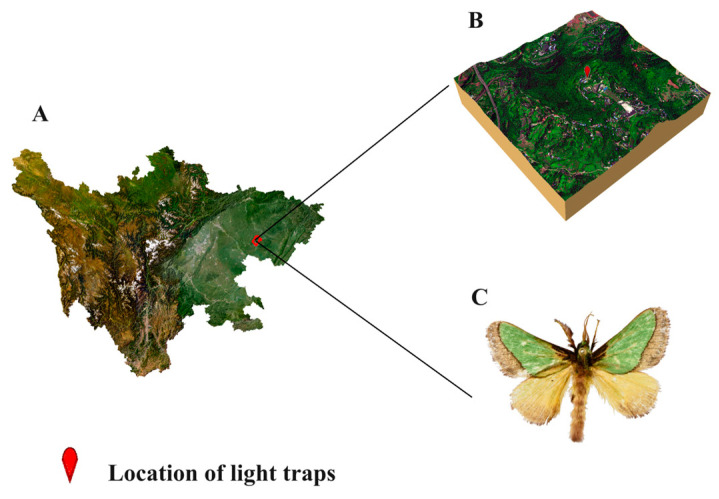
Adult of *L. consocia* and its location of collection. (**A**) Relief map of Sichuan Province, China. (**B**) Relief map of the collected location. (**C**) Adult of *L. consocia*. The specimen of *L. consocia* were collected by light traps from Xishan Forest Park, Nanchong City, Sichuan Province, China, on 23 July 2023 (30.805692° N, 106.054003° E, Alt. 363 m), and were deposited in the Laboratory of Forest Conservation, College of Life Science, China West Normal University (Voucher No. SCNC-LCW-20230723.1). The photo was taken by Yaqing Peng with Canon EF 180 mm f/3.5 L USM (Canon Inc., Tokyo, Japan).

**Figure 2 insects-15-00756-f002:**
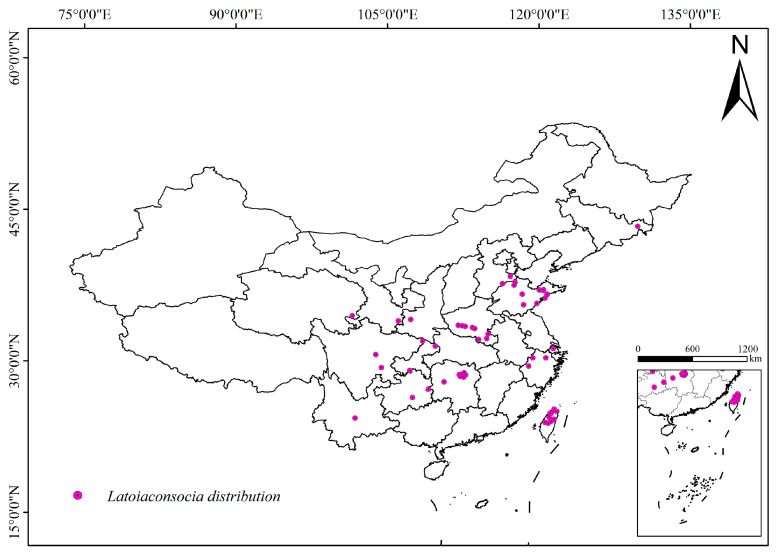
Occurrence records of *L. consocia* in China.

**Figure 3 insects-15-00756-f003:**
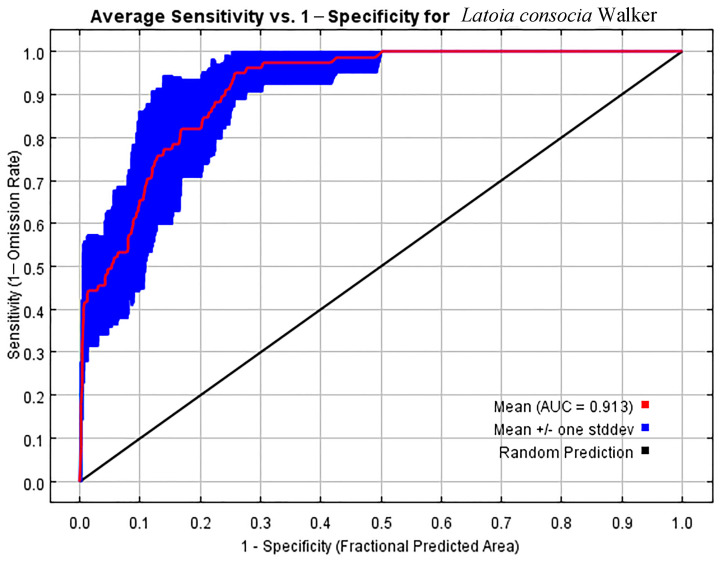
ROC curve and AUC value. ROC represents receiver operating characteristic, and AUC represents area under the ROC curve.

**Figure 4 insects-15-00756-f004:**
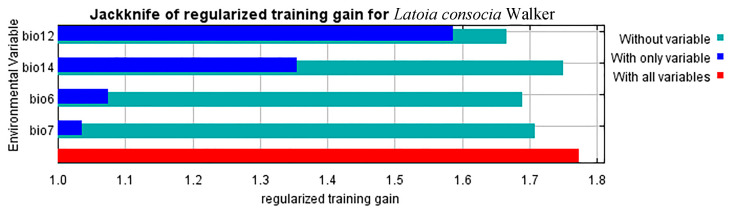
Variable importance determined via the folding jackknife test.

**Figure 5 insects-15-00756-f005:**
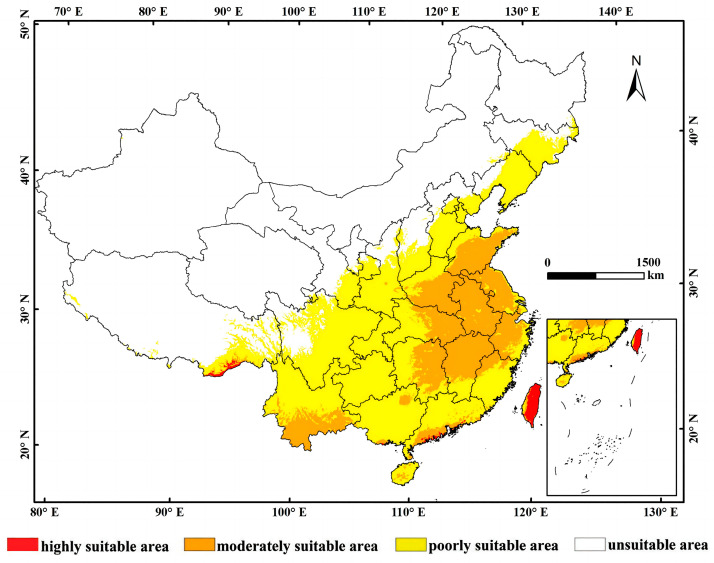
Current distribution of suitable habitats for *L. consocia.* The color blocks of the region show the probability of the occurrence of *L. consocia*. Red represents a probability higher than 0.66 of being a high suitability area, orange represents a medium suitability area with a 0.33–0.66 probability, yellow represents low suitability areas with a 0.05–0.33 probability, and white represents unsuitable areas.

**Figure 6 insects-15-00756-f006:**
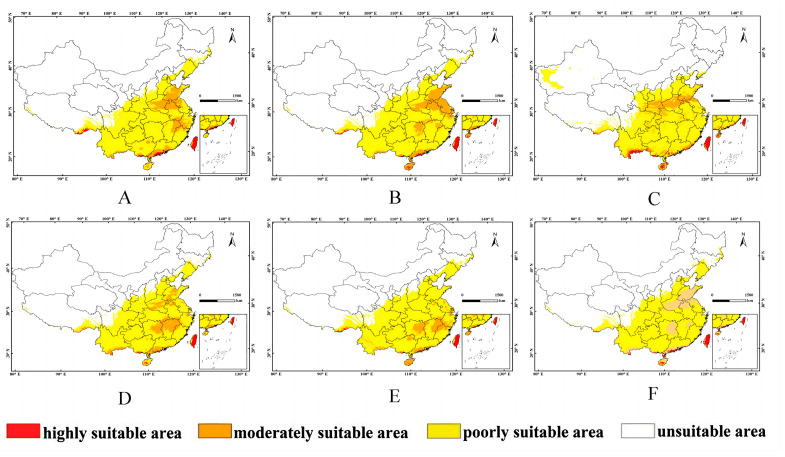
The potential distribution of *L. consocia* in suitable regions in China under different climatic conditions. The color blocks of the region show the probability of the occurrence of *L. consocia*. Red represents a probability higher than 0.66 of being a high suitability area, orange represents a medium suitability area with a 0.33–0.66 probability, yellow represents low suitability areas with a 0.05–0.33 probability, and white represents unsuitable areas. (**A**) stands for 2050s, SSP1-2.6, (**B**) stands for 2050s, SSP2-4.5, (**C**) stands for 2050s, SSP5-8.5, (**D**) stands for 2090s, SSP1-2.6, (**E**) stands for 2090s, SSP2-4.5, and (**F**) stands for 2090s, SSP5-8.5.

**Figure 7 insects-15-00756-f007:**
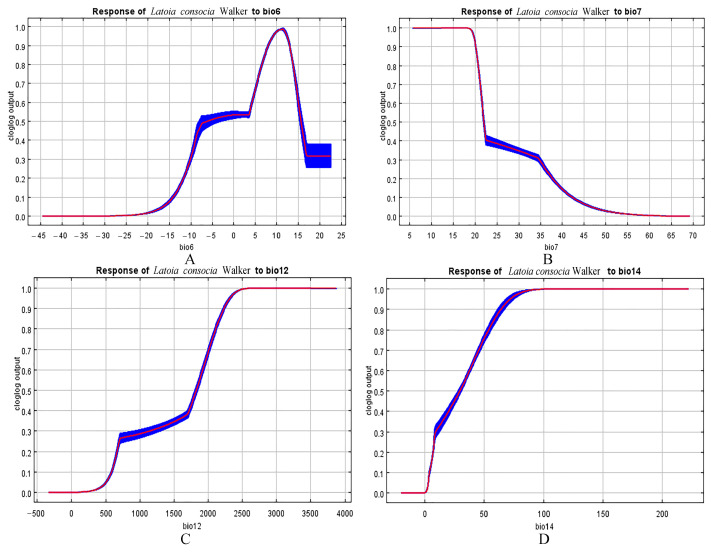
Response curves of key environmental variables. (**A**) Minimum temperature of the coldest month (Bio6), (**B**) temperature annual range (bio5–bio6) (Bio7), (**C**) annual precipitation (Bio12), and (**D**) precipitation of the driest month (Bio14). The curves are represented by the mean (red line) and standard deviation (SD, blue shaded area).

**Table 1 insects-15-00756-t001:** Pearson correlation coefficients of key environmental factors.

Code	bio6	bio7	bio12
bio7	−0.069		
bio12	0.695 **	0.531 **	
bio14	0.648 **	0.333 **	0.845 **

**, significant correlation at the 0.01 level (bilateral).

**Table 2 insects-15-00756-t002:** Percentage contribution and permutation importance of environmental variables in the MaxEnt model.

Variable	Environment Variables	Percent Contribution (%)	Permutation Importance (%)
Bio 12	Annual precipitation	74.8	27.75
Bio 14	Precipitation of driest month	13.41	8.56
Bio 6	Min. temperature of coldest month	9.22	53.12
Bio 7	Temperature annual range	2.57	10.58

**Table 3 insects-15-00756-t003:** Analysis of the main suitable distribution areas for *L. consocia*.

Province	Highly Suitable Area (km^2^)	Total (10^4^ km^2^) *	Percentage of Highly Suitable Areas in Province (%)
Taiwan	1450.00	3.60	4.028
Tibet	472.00	120.28	0.039
Guangdong	261.00	17.98	0.145
Hong Kong	33.00	0.11	3.000
Guangxi	23.00	23.76	0.010
Yunnan	3.00	39.41	0.001
Hainan	1.00	3.54	0.003
China	2243.00	960.00	/

* Indicates the total area of the corresponding province.

**Table 4 insects-15-00756-t004:** Prediction of the suitable areas for *L. consocia* under current and future climatic conditions.

Decade Scenarios		Predicted Area (10^4^ km^2^)			Comparison with Current Distribution (%)		
		Poorly Suitable Aera	Moderately Suitable Aera	Highly Suitable Aera	Poorly Suitable Aera	Moderately Suitable Aera	Highly Suitable Aera
Current		35.78	19.39	0.22			
2050s	SSP1-2.6	36.33	18.75	0.31	1.55	−3.31	39.32
	SSP2-4.5	36.51	18.63	0.16	2.03	−3.89	−26.48
	SSP5-8.5	35.72	19.20	0.36	−0.18	−0.95	60.41
2090s	SSP1-2.6	36.81	18.39	0.20	2.88	−5.16	−13.02
	SSP2-4.5	35.89	19.29	0.22	0.31	−0.53	−3.52
	SSP5-8.5	36.78	18.31	0.30	2.79	−5.55	34.20

## Data Availability

The original contributions presented in the study are included in the article; further inquiries can be directed to the corresponding author.

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
