# Peer review of "Mapping Species Distributions of *Latoia consocia* Walker under Climate Change Using Current Geographical Presence Data and MAXENT (CMIP 6)"

_insects, 2024, doi:10.3390/insects15100756_

Round 1

Reviewer 1 Report

Comments and Suggestions for Authors

General Comments:

This manuscript presents a modeling effort to predict the potential future distribution and environmental factors determining the geographic distribution of Latoia consocia. The authors use a commonly used modeling approach (MAXENT) to determine the range of L. consocia and then use it to predict the potential future range based on climate change predictions. The manuscript is generally well written, but is lacking some key information regarding the biology of the organism that makes interpreting the validity of the methods and results a little difficult. Please see below

Major comments:

1)    I did not have much experience with the biology of L. consocia, which made me uncertain as to whether a range prediction based on climatic factors would be useful to predict the future distribution of the species. For many Lepidopteran species the range of its hosts is often the primary determinant of its distribution and climate may simply determine host locations. Therefore, predicting future spread based on climate may not be completely appropriate. The authors state that the species uses many hosts, but it is never stated what these are and no clear reasoning indicating that hosts are commonly found in areas where this species is not located was given (i.e. an indication the climate is what restricts the range). Unfortunately, when I attempted to find information on this insect I could not find that much information on its host use or as a major forest/agricultural pest. Most information of its pest status seemed to concern the effect of its toxins on humans. However, I found information that the name may be a synonym for Parasa consocia, which I found confusing as P. consocia has a much greater distribution than implied in the manuscript. Please provide more information regarding the ecology of the organism.

2)     The authors separated the data set into a training (75%) and testing (25%) components which is very reasonable. However, from the manuscript it appears that points were selected at random, which if they were I would expect a map tested on points adjacent to training points or from the same areas would be expected to have a great fit. Testing data should ideally be from areas not used in the training, not points within the same areas. If the species is indeed Parasa consocia and is found in areas outside of those used to train the model, those areas would seem ideal for testing.  

3)    If Latoia consocia and Parasa consocia are the same species, why was the greater distribution of the species not considered? I assume that they are the same species as a paper cited (# 9) names the species Parasa consocia, unfortunately I was not able to locate the paper.

4)    AUC has often been criticized as an insufficient measure of model quality. Please consider including other estimates in addition to AUC. Please see the manuscript below and works citing them subsequently,

Jiménez‐Valverde, A., 2012. Insights into the area under the receiver operating characteristic curve (AUC) as a discrimination measure in species distribution modelling. Global Ecology and Biogeography, 21(4), pp.498-507.

Minor comments:

1. Abstract lines 16-17: The authors state that this pest has rapidly spread. However, where or when is never mentioned again. It also states that poses a threat to related plants, but which plants it consumes is never mentioned.

2. Introduction and Discussion: The introduction as a whole discusses the MAXENT approach in a fair amount of detail. However, MAXENT is a fairly well known approach and greater emphasis on the biology of the species and purpose of the current study rather than MAXENT would be appreciated.

Reviewer 2 Report

Comments and Suggestions for Authors

The presented manuscript deals with predicting the current and future global suitability of Latoia consocia, pest causing damage to fruit trees, forest trees, and other vegetation. The authors have collected a reasonable amount of data, nevertheless there are a number of issues concerning the contend of the introduction and applied methodology, as follows:

Introduction:

1) lines: 47- 54, the authors are writing about habitat suitability models for plant, whethouth such need, since the paper is on habitat suitability models for an insect pest.

2) line 69: “While most larvae of the larvae of the foliar pest”, should be “While most larvae of the foliar pest” (or a most specific name for the group the target insect belongs to.

3) “The foliar pest mainly feeds”, what do you mean by “foliar pest”, is it a common name for the family Limacodidae? Be more specific, since it is unclear about what species or group of species are the next few sentences, as well.

4) “The foliar pest not only impact human health…”, may a short explanation of what impact consists of would be useful.

Materials and Methods.

1)      A major problem that you evaluate the model performance using metrics (namely, AUC) designed for analyses with reliable absence information (e.g.), under the misconception that they represent statistically unbiased, absolute measures of performance, see “Lobo, J. M., Jiménez-Valverde, A. and Real, R. 2008. AUC: a mis leading measure of the performance of predictive distribution models. – Global Ecol. Biogeogr. 17: 145–151.”, and also: Leroy, B., Delsol, R., Hugueny, B., Meynard, C. N., Barhoumi, C., Barbet-Massin, M. and Bellard, C. 2018. Without quality presence–absence data, discrimination metrics such as TSS can be misleading measures of model performance. – J. Biogeogr. 45: 1994–2002. Therefore, I strongly suggest the use of adjustments for differential weighting of omission and commission errors or other ways to meliorate the evaluation problems see SoleyGuardia et al (cited above).

2)      lines 118-119, “The occurrence data of L. consocia used in this study were obtained through the following methods, Global Biodiversity Information Facility (GBIF).”, GBIF is not the method, please correct.

3)      “Figure 1 shows the morphology of the L. consocia collected from different regions.”, since you are not commenting on any morphological characters of the species, it is better to change “morphology” to habitus.

4)      “The distribution map of L. consocia can be seen in Figure. 2.”, change “distribution” to occurrence.

5)      “The figure was taken by Yaqing Peng”, change “figure” to “photo”.

6)      Correct the sentence “This research from Global WorldClim database (http://www.worldclim.org/download) to download the 19 historical Global biological climate variables.”, It is not clear, some words are missing.

7)      Line 152: “These bioclimatic variables closely affect the growth and development of L. consocia.”, please provide reference.

8)      “Climatic variables with correlation coefficients greater than 0.85 were removed to reduce the influence of overfitting on the model, ensuring the robustness and accuracy of the model (Xu et al. 2019)”, change the format of the citation, add the reference information to the references list;  and add at least one paper on methodology of controlling for collinearity, not just some one applying the chosen by you specific threshold of correlation coefficients.

9)  Line 171-173: “The receiver operating characteristic (ROC) curve and the area under the ROC curve (AUC) are commonly used methods to assess model accuracy. The AUC value ranges from 0 to 1” remove, you have said that already - lines 102-103.

10) “According to the IPCC report on probability classification methods” add the full reference to the list of references, and a citation with the appropriate formatting here.

Kind regards

Round 2

Reviewer 1 Report

Comments and Suggestions for Authors

Comments on revision : In the current submission the authors have made a great job of addressing the comments provided in the previous draft. I only have a few minor comments.

1) The sections added in response to reviewer comments could be edited for clarity and grammar. The grammatical errors are minor, but a small amount of editing would be appreciated to match the quality of the rest of the mansucript.

Comments on the Quality of English Language

The majority of the manuscript is written very well. However, the sections added as part of the revision need some minor editing for grammatical errors.

Reviewer 2 Report

Comments and Suggestions for Authors

Thank you for considering my suggestions. I still have a few recommendations: 

1)      I found some discrepancies in Table 4., please check all the numbers and correct carefully.

2)      Line 355 “The MaxEnt model predicted that different species have different potential distribution-suitability areas.” – it does not mean anything, remove it, please.

3)      Lines 386 – 419, reduce these paragraphs, do not repeat the results and remove all statements which are too general, or try to show what is the connection with your study.

Kind regards
